# Correcting Coherent Errors by Random Operation on Actual Quantum Hardware

**DOI:** 10.3390/e25020324

**Published:** 2023-02-10

**Authors:** Gabriele Cenedese, Giuliano Benenti, Maria Bondani

**Affiliations:** 1Center for Nonlinear and Complex Systems, Dipartimento di Scienza e Alta Tecnologia, Università degli Studi dell’Insubria, Via Valleggio 11, 22100 Como, Italy; 2Istituto Nazionale di Fisica Nucleare, Sezione di Milano, Via Celoria 16, 20133 Milano, Italy; 3NEST, Istituto Nanoscienze-CNR, 56126 Pisa, Italy; 4Istituto di Fotonica e Nanotecnologie, Consiglio Nazionale delle Ricerche, Via Valleggio 11, 22100 Como, Italy

**Keywords:** quantum computing, NISQ devices, quantum error correction, random quantum circuits

## Abstract

Characterizing and mitigating errors in current noisy intermediate-scale devices is important to improve the performance of the next generation of quantum hardware. To investigate the importance of the different noise mechanisms affecting quantum computation, we performed a full quantum process tomography of single qubits in a real quantum processor in which echo experiments are implemented. In addition to the sources of error already included in the standard models, the obtained results show the dominant role of coherent errors, which we practically corrected by inserting random single-qubit unitaries in the quantum circuit, significantly increasing the circuit length over which quantum computations on actual quantum hardware produce reliable results.

## 1. Introduction

Quantum computers operating with ∼50–100 qubits may be able to perform tasks surpassing the abilities of the present classical digital super-computers [1,2], and have recently been suggested to possess a quantum advantage in specific problems [3,4,5]. However, this quantum advantage can only be reached with a high enough quantum gate precision and through-processes that generate enough entanglement to outperform the classical tensor network methods [6]. Unfortunately, noisy intermediate-scale quantum (NISQ) devices at present suffer from significant decoherence and the effects of various sources of noise, such as residual inter-qubit coupling or interactions with an uncontrolled environment. Noise limits the size of the quantum circuits that can be reliably executed; therefore, achieving a quantum advantage in complex, practically relevant problems is still an imposing challenge.

It is, therefore, important to benchmark the progress in the currently available quantum computers [7,8], and possibly find suitable error-mitigation strategies. Here, we focus on freely available IBM quantum processors. IBM provides a few characteristic noise parameters for these processors: qubit relaxation time, qubit dephasing time, and error rates in single-qubit gates, two-qubit CNOT gates and quantum measurements. Such parameters are updated after each hardware calibration and provide very useful information for the assessment of quantum hardware performance. Nevertheless, as we will show below, these noise channels are not sufficient for an accurate description of the errors affecting a quantum computer. As a general observation, we note that, even in the simplest case of memoryless errors, the general description of quantum noise in terms of quantum operations requires many real parameters $N_{p}=N^{4}-N^{2}$, where $N=2^{n}$ is the Hilbert space dimension for *n* qubits [1]. For a single qubit $N_{p}=12$, these parameters have a simple intuitive interpretation in terms of rotations, deformations, and displacements of the Bloch sphere. Furthermore, coherent errors [9,10,11,12,13,14,15,16,17,18], that is, unitary errors that slowly vary relative to the gate time, can arise for several reasons, including miscalibration or drifts away from the control system calibration used to drive the qubit operations, cross-talk with neighboring qubits, external fields, and residual qubit–qubit interactions. Such errors cannot be removed with standard error-correcting codes [1] developed for stochastic (incoherent), uncorrelated, memoryless errors.

Here, we describe the evolution of a qubit inside an operating quantum computer as a quantum operation or a completely positive trace-preserving (CPT) map acting on the single-qubit Bloch sphere. Our first goal is to provide a full characterization of such a CPT map, that is, a *quantum process tomography* going beyond the few noise channels noted above, in order to analyze the performances of the current noisy quantum hardware and define a useful benchmark for new releases of quantum computers. We consider a *quantum echo experiment* as the quantum noise travel, reversing a quantum computation, so that in the ideal noiseless case we could reconstruct the initial state. More specifically, we consider an even sequence of CNOT gates (${\mathrm{CNOT}}^{2}=I$) and a more general sequence of two-qubit random unitary operators, combined with their time-reversal operators, to perform the echo experiment. The comparison between the results from the quantum noise tomography on the IBM quantum hardware and those obtained with the Qiskit simulator, which only considers a limited number of noise channels, highlights the relevance of coherent errors. This observation suggests that random single-qubit gates could be introduced to the logical circuit to suppress coherent errors [19,20]. We demonstrate the effectiveness of such a strategy in the above CNOT–echo experiment. Alternatively, in the case of a realistic quantum computation without echoes, one could insert random Pauli gates between circuit elements, such as CNOTs (randomized compiling), as shown in [20,21,22].

The paper is organized as follows. In Section 2 we recall the Bloch sphere representation of a CPT map for a single qubit and the main steps in the consequent quantum process tomography. In Section 3, the echo results obtained from IBM quantum processors are shown and compared with those of the IBM simulator. In Section 4 we discuss the effectiveness of the randomization strategy for coherent error correction. Our conclusions are drawn in Section 5.

## 2. CPT Maps on the Bloch Sphere

As it is well-known, a two-level system (a qubit) state can be represented by a point in a ball of unit radius, called the Bloch ball, embedded in $R^{3}$, which defines the so-called Bloch vector (r). Pure states are those that lie on the surface of the ball (|r|=1), i.e., the Bloch sphere, while mixed states are those inside the sphere (|r|<1). An ideal quantum echo experiment should leave pure states on the sphere, while, in the real case, the length of the Bloch vector generally decreases.

Given an initial single-qubit state ρ, we considered the quantum noise channel, or a completely positive trace-preserving (CPT) map S, as a quantum black box, representing the qubit interacting with a generic physical system. Without any a-priori knowledge of the quantum noise processes affecting the qubit, we could reconstruct S by preparing different input states ρ, with each of them measuring the output state $\rho^{\prime }=S\left(\rho \right)$. In the Bloch-sphere representation, the Bloch vector evolves as an affine map: (1)$$r\to r^{\prime }=Mr+c.$$

To perform a full quantum process tomography, i.e., to reconstruct the matrix *M* and the vector c, we needed to perform 12 different experiments. In each experiment, we prepared one of the four initial states
(2)$$|0\rangle ,\;\;|1\rangle ,\;\;|x\rangle =\frac{1}{\sqrt{2}}\left(|0\rangle +|1\rangle \right),\;\;|y\rangle =\frac{1}{\sqrt{2}}\left(|0\rangle +i|1\rangle \right).$$

For each initial state, we measured the polarization of the final state $\rho^{\prime }$ down the channel, along one of the three coordinate axes, to estimate the final Bloch vector for each input state. Each experiment was repeated many times, $N_{r}=25$, with $N_{m}=8192$ runs each time, to reconstruct *M* and c with high accuracy. In the case of a sequence of random unitaries, $N_{r}=25$ sequences of two-qubit random unitaries were extracted from the Haar measure on the unitary group U(4) [23,24,25].

The quality of the quantum channel was measured by the fidelity F between the ideally pure single-qubit initial state $|\psi_{in}\rangle$ and the final, generally mixed, state $\rho_{out}$: (3)$$F=\langle \psi_{in}|\rho_{out}|\psi_{in}\rangle .$$

We evaluated the fidelity using both the Qiskit simulator and the real quantum hardware for the initial states of Equation (Equation 2) and the echo protocols described below.

## 3. Single-Qubit Quantum Process Tomography

All experiments on actual quantum hardware were performed on ibm_lagos. We reconstructed the evolution of the Bloch sphere of qubit 0 ($q_{0}$) with qubit 1 ($q_{1}$) as the ancilla. The CNOT gates of the CNOT noisy channel were performed with $q_{0}$ as the control qubit. Note that using $q_{0}$ as the target qubit did not qualitatively alter the results.

### 3.1. CNOT Noisy Channel

The fidelities obtained with the sequence of noisy CNOTs (the CNOT noisy channel) are shown in Figure 1. The results of the Qiskit simulations are easily interpreted, as follows. Since one problem in the CNOT is the long gate time, the dephasing and the energy relaxation errors (parametrized by the decoherence times $T_{1}$ and $T_{2}$) become important. The state |0〉, which is not affected by these kinds of error, presents better fidelities than |1〉, |*x*〉 and |*y*〉; with |1〉 being slightly better than |*x*〉 and |*y*〉, since it is only affected by the relaxation. The results obtained with the actual device did not reflect the Qiskit simulation, and the reason for this can be understood by looking at the evolution of the Bloch sphere depicted in Figure 2. In the Qiskit simulations, the sphere deforms, becoming an ellipsoid with a semi-major axis at the *z*-axis, and the center shifts in the direction of positive *z*’s. This mirrors what occurs with fidelities, with states near the north pole (which represent the state |0〉) of the spheres being gradually less affected by dephasing and energy relaxation. Ellipsoids are still formed in the real quantum hardware case, but they appear to be rotated, with the semi-major axis no longer in the *z*-direction. This can be interpreted (see below) as the result of some kind of coherent error, which induces undesired rotations of the Bloch sphere.

### 3.2. Random Unitaries

We now consider sequences of two-qubit random unitaries $U_{k}$, with *k* running from 1 to the number of steps $N_{s}$, combined with their inverse $U_{k}^{†}$, to realize the echo experiment. In summary, we implemented the overall random quantum circuit $\prod_{k=1}^{N_{s}}U_{k}^{†}U_{k}$. In contrast with the CNOT channel, as shown in Figure 3, the Qiskit simulation’s fidelities of the four basis states are comparable. This can be easily understood: the CNOTs in the decomposition into elementary quantum logic gates of general two-qubit unitary operators are always preceded by a single-qubit random rotation (see Appendix A). As a consequence, whatever the initial state, CNOTs act on a random state of the sphere, and the error due to dephasing and relaxation is, on average, independent of the initial state. For the same reason, we anticipate that the effects of coherent errors cancel each other out. Indeed, we expect from the previous literature [19] that the deleterious effect of such errors is greatly reduced if they are randomized by repeatedly rotating the computational basis via random single-qubit unitaries. The fact that, unlike the CNOT channel, the actual hardware has a comparable performance to that predicted by the Qiskit simulator (see Figure 4), demonstrates that, for the CNOT channel, hardware peformance degradation should largely be ascribed to coherent errors.

## 4. Error Correction by Randomization

In the case of the CNOT noisy channel, the nature of the coherent noise suggests a natural error correction procedure that occurs repeatedly rotating the computational basis. The idea is that, before any CNOT pair, a random single qubit unitary gate is applied at each qubit of the noise channel and the respective adjoint after the CNOTs to globally obtain the identity. Alternatively, we took randomly chosen rotations with respect to x or y (also randomly selected) to decrease the number of logic gates used for the correction (see Appendix B). As we can see in Figure 5, the fidelities significantly improve when compared with those of the CNOT noisy channel without correction; moreover, they are independent of the initial state. The results with single-axis rotations are comparable with those of generic single-qubit unitary operators.

However, in a generic quantum computation, the circuit is not reversed as it is for a CNOT pair, for which ${\left(\mathrm{CNOT}\right)}^{2}=I$. Nevertheless, coherent errors can still be mitigated. For instance, a single CNOT gate can be treated by compensating the pre-CNOT random rotations with suitable rotations after the CNOT so that the CNOT gate is effectively operated overall. To achieve this purpose, we exploited the commutation rules between CNOT and single-qubit rotations. Note that, instead of a generic rotation, it is easier and sufficient to consider randomly chosen Pauli gates {I,X,Y,Z} [20,21,22], for which we have the following identities: (X⊗X)CNOT(X⊗I)=CNOT, (Y⊗Y)CNOT(X⊗(−Z))=CNOT, (X⊗Y)CNOT(X⊗(−Y))=CNOT, and so on.

## 5. Conclusions

We have noted the presence of coherent errors in echo experiments performed on actual quantum hardware. We have considered two quantum noise channels: one formed by a sequence of CNOT gates, the other by random two-qubit interactions. The former clearly shows the presence of coherent errors by observing the rotation of Bloch spheres. The latter suggests that randomization of the computation basis leads to the natural cancellation of coherent errors, as we practically demonstrate for a sequence of CNOT gates. Removing this kind of noise is the first step in implementing a successful error correction procedure.

## Figures and Tables

**Figure 1 entropy-25-00324-f001:**
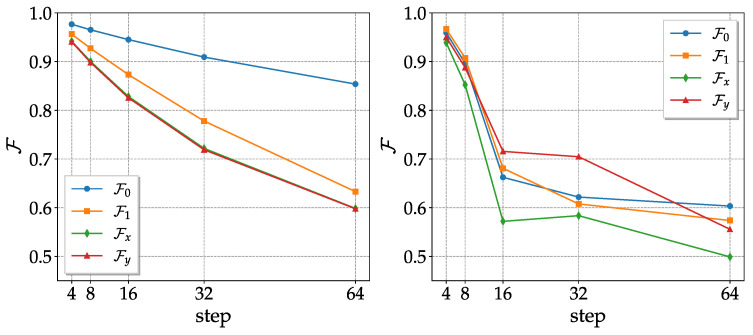
Fidelities of the 4 states used to reconstruct the CPT map as a function of the number of steps (one step of the CNOT noisy channel corresponds to two CNOT gates). (**Left**) simulations with the noise parameters of *ibm_lagos*, calibration of 2 November 2022. (**Right**) actual results obtained with *ibm_lagos* on 2 November 2022.

**Figure 2 entropy-25-00324-f002:**
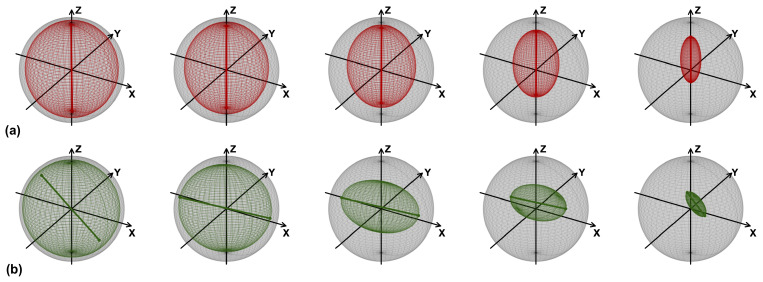
Evolution of the single qubit Bloch ball as a function of the number of CNOT map steps. From left to right, the number of steps increases in correspondence with the data shown in Figure 1. (**a**) Qiskit Simulations with the noise parameters of *ibm_lagos*. The segment highlighted in red shows the major axis of the ellipsoid; the gray sphere is the unit-radius Bloch ball. (**b**) Results obtained with *ibm_lagos*. The segment highlighted in green shows the major axis of the ellipsoid; the gray sphere is unit-radius Bloch ball. Data from the quantum processor, taken on 2 November 2022, with the corresponding calibration parameters used for Qiskit simulations.

**Figure 3 entropy-25-00324-f003:**
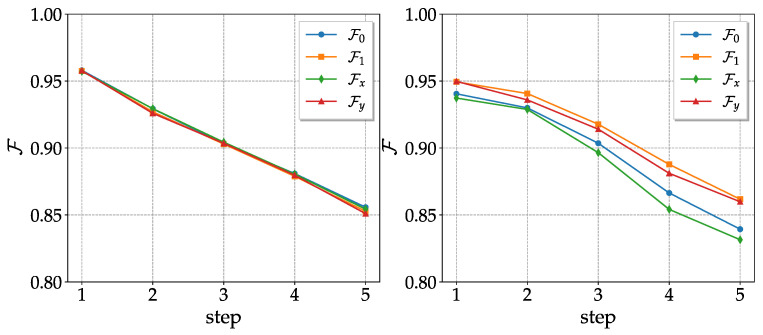
As in Figure 1, but for random unitaries. Qiskit (**left**) and actual hardware (**right**) data were obtained with *ibm_lagos* on 2 November 2022.

**Figure 4 entropy-25-00324-f004:**
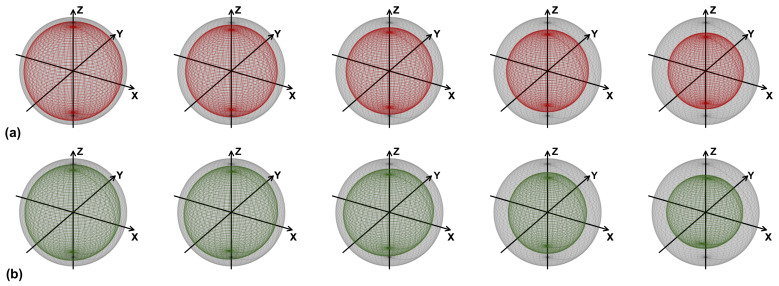
As in Figure 2, but for the random unitaries channel of Figure 3.

**Figure 5 entropy-25-00324-f005:**
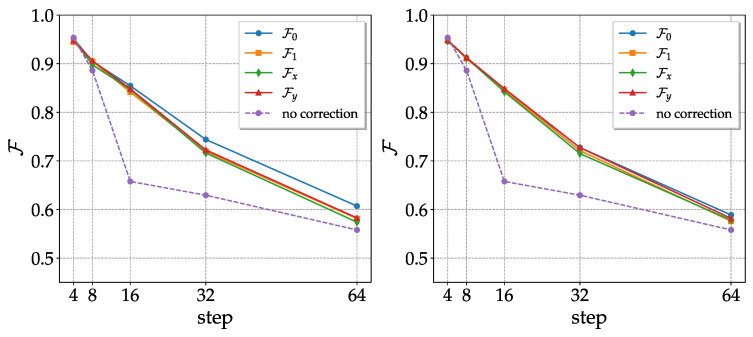
Fidelities of the 4 basis states as a function of the number of steps (one step in the CNOT noisy channel corresponds to two CNOT gates) after the correction procedure described above. On the left, the computational basis is rotated with random unitaries; that on the right undergoes single-axis rotations. The purple curve represents the average fidelities obtained without correction. Results obtained with ibm_lagos on 20 November 2022.

## Data Availability

The datasets used and analyzed in the current study are available from the corresponding author on reasonable request.

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
