# Peer review of "Correcting Coherent Errors by Random Operation on Actual Quantum Hardware"

_entropy, 2023, doi:10.3390/e25020324_

Round 1
Reviewer 1 Report
My main concern with this paper is that it is not very original.
There is a growing and now mature literature on using randomization methods to suppress coherent errors via pauli frame randomization and randomized compiling and the authors seems to be unaware of these prior results.
This literature includes both theoretical and experimental results. Unfortunately the authors are here “rediscovering” ideas that are already well developed and do not seem to realize this and do not seem to be addressing current challenges that have grown from the existing literature.
The work could be of interest if the authors connected their approach to that existing literature. How is it different and unique and teaching us something new? Without addressing this point I can not recommend this paper for publication.
Author Response
Thank you for your comments. Please find our replies attached.

Reviewer 2 Report
The paper deals with error mitigation of near-term quantum hardware. The authors perform full quantum process tomography of single qubit operations on a real quantum computer. They adapt quantum echo experiments for that purpose. The authors attempt to correct the errors by inserting random single-qubit unitaries to the quantum circuit.
Error mitigation is of high practical importance. Noisy quantum computers are unlikely to achieve advantage over classical computers without accurate and reliable error mitigation. New techniques leading to improvements in our current ability to mitigate errors on quantum computers are therefore highly sought after.
The paper is technically correct but it lacks novelty. It makes very simple observation and implements error mitigation technique that has been presented long time ago at the smallest possible scale. The paper does not push the field forward. It does not demonstrate new capability (like correcting deeper circuits) and does not include ideas for new, useful benchmarks or analysis methodologies. Therefore, I do not recommend publication. I provide more detailed reasoning below.
The authors concentrate on publicly available IBM quantum computers. IBM provides error rates after each calibration, so the researchers have an idea of the strength of noise that affects a given qubit. The authors state that the information provided is not sufficient for accurate description of errors. This is very well-known fact and all the groups I have interacted with perform their own full gate tomography to build their error models. They typically use well-established, publicly available software.
The authors observed that the fidelities of CNOT noisy channel obtained with Qiskit noisy simulator do not agree with measured ones (Fig. 1). However, this discrepancy goes away for random unitaries. This suggests that the choice of initial states in Eq. (2) is biased. Indeed, the device is not "symmetric" under the exchange of |0> and |1>. In other words, the choice of states in Eq. (2) is not representative. A more refined error model would take that into account and the simplistic one provided by IBM after each calibration does not do that. This is actually reasonable, since full characterization is expensive and it is better to let users work on a device rather than run long, accurate metric analysis that most users do not need.
The authors turn the observation (random performs better than CNOT) into error mitigation strategy. Their idea is to insert random single-qubit unitary without changing the outcome of the experiment. This method is not new, the authors cite the paper from 2005. Since then, that method has been extended and became a practical error mitigation tool taking many shapes and forms.
Author Response
Thank you very much for your comments. Please find our replies attached.

Reviewer 3 Report
The study concerns an important question of what types of errors do actually take place in hardware. The authors underline the role of coherent errors by clearly demonstrating the discrepancy between the mathematical simulation (T1-T2 model) and the experimental tomography. The proposed method to fight the coherent errors is to exploit randomization. Introduction and section 4 should be extended in this regard to discuss quantum randomized compiling. The description of the methodology in Section 4 should be clarified too, namely, the authors write "in order to globally obtain the identity", however, if one does not invert the circuit, the goal is still to obtain the CNOT gate (not identity). How does the pre-CNOT random rotation should be compensated after CNOT in this case?
Author Response
Dear Editor,
we thank the Reviewers for their careful reading of the manuscript and for the questions and suggestions. We have revised our manuscript accordingly.
Below, you can find the answers to the comments, reproduced in italics, and a list of changes. We hope that the revised manuscript is now suitable for publication.
Sincerely,
Gabriele Cenedese, Giuliano Benenti and Maria Bondani
Reviewer 1
Reviewer: Introduction and section 4 should be extended in this regard to discuss quantum randomized compiling.
Answer: We extended the introduction and section 4 by including a comment on randomized compiling and added some significant references.
Reviewer: The description of the methodology in Section 4 should be clarified too, namely, the authors write "in order to globally obtain the identity", however, if one does not invert the circuit, the goal is still to obtain the CNOT gate (not identity). How does the pre-CNOT random rotation should be compensated after CNOT in this case?
Answer: Regarding the methodology in Section 4, we have added the quantum circuit of the CNOT noisy channel in the Appendix B. We have added a brief comment on the strategy to be used to compensate random rotations in the case of no echo, that is for a generic protocol.
Reviewer 2
We thank the Reviewer for his/her appreciation.
List of changes (all changes are highlighted in blue in the manuscript):
- In the Introduction, we added a brief comment on randomized compiling;
- In Section 4 we added a brief comment on random rotations compensation;
- We added appendix B;
- We added some new References.
Reviewer 4 Report
Correcting Coherent Errors by Random Operation on Actual
Quantum Hardware
Gabriele Cenedese, Giuliano Benenti and Maria Bondani
With existing noisy quantum devices, error mitigation is an important challenge. In this paper, the authors consider coherent errors in full quantum process tomography of single qubit in an IBM machine which they could correect by inseritng random single-qubit unitaries in the quantum circuit. The results are interesting and I would recommend publication in its current form.
Author Response

(The authors gave the same response as above.)

Round 2
Reviewer 1 Report
The authors’ response is not satisfactory. This work may have been relevant 20 years ago but is irrelevant now. the paper has no novelty and should not be published. For example, single qubit process tomography has been demonstrated over 20 years ago. There is a vast literature on the topics addressed in the paper that the authors have neglected.Reviewer 2 Report
The authors have modified their manuscript but not substantially. Their response is not sufficient to change my assessment.
The authors should check pyGSTi software (https://www.pygsti.info/). This is the software that I was referring to in my first review. ArXiv:2002.12476 describes the package in detail. The paper has been cited 50 times and the packaged has been downloaded almost 200 times. It should not be hard to find researchers using it in everyday work.
I agree, the paper has some pedagogical aspects. Is that enough for the paper to be published in Entropy? I do not think so, but I ask the editor to make the final decision. I believe that the fact the paper “may stimulate further investigations” is simply not enough to warrant the publication in a reputable journal. I still believe the paper is not sufficiently innovative to be published as a regular research article.